# Poor Dietary Polyphenol Intake in Childhood Cancer Patients

**DOI:** 10.3390/nu11112835

**Published:** 2019-11-19

**Authors:** Ashly Liu, Jennifer Cohen, Orazio Vittorio

**Affiliations:** 1UNSW Medicine, University of NSW, Randwick, NSW 2031, Australia; ashlyliu21@gmail.com; 2School of Women’s & Children’s Health, UNSW Medicine, University of NSW, Randwick, NSW 2031, Australia; ovittorio@ccia.org.au; 3Kids Cancer Centre, Sydney Children’s Hospital, Randwick, NSW 2031, Australia; 4Children’s Cancer Institute, Lowy Cancer Research Centre, UNSW Sydney, Randwick, NSW 2031, Australia

**Keywords:** childhood cancer, polyphenols, dietary intake, diet quality, nutrition

## Abstract

Emerging research demonstrates polyphenol-rich diets like the Mediterranean diet may play a role in improving the outcomes of adult cancer therapy. To date, there are no trials assessing the intake or efficacy of polyphenol-rich diets in childhood cancer patients. In this study we collected dietary data on 59 childhood cancer patients on treatment using a three-pass 24-h dietary recall (24-HDR), which is based on a validated and structured three-part methodology. Polyphenol consumption was calculated by matching the food consumption data with polyphenol content extracted from the most updated Phenol-Explorer database. The mean total polyphenol intake was 173.31 ± 141.02 mg/day. The major food sources of polyphenols were fruits, beverages, and cereals. There were no significant associations with time since diagnosis, body mass index (BMI) z-score, types of cancer, treatment intensity, food-related symptoms, relapse, and total daily polyphenol intake. Further investigation with larger studies will facilitate the steps in assessing the value of polyphenol-rich dietary patterns in future nutritional interventions for childhood cancer patients.

## 1. Introduction

Poor nutritional status and malnutrition during and following paediatric cancer treatment have been linked with an increase in adverse outcomes of higher relapse and lower survival rates [1]. Subsequently, research into childhood cancer therapies is starting to focus on novel strategies to improve their efficacy and further improve survival outcomes in childhood cancer patients [2,3]. As a healthy diet has long been encouraged in improving the outcomes of adult cancer care, there is an opportunity amongst clinicians and researchers to investigate effective dietary changes to improve survival in childhood cancer patients [2]. There is now an acknowledgement of the need for better nutritional interventions throughout cancer management in light of their potential to enhance patient survival associated with cancer treatment [4,5,6]. 

It has long been understood that a regular consumption of healthy dietary patterns, such as the Mediterranean diet, is associated with a reduced risk of cancer [7]. The reduced development of cancer cells has been attributed to the presence of antioxidants, primarily polyphenols, which are found in fruits, vegetables, and olive oil that are characteristic of a traditional Mediterranean diet [8]. Although it is difficult to conduct blinded dietary intervention studies, there have been systematic reviews and meta-analyses of the adherence to healthy dietary patterns, conveying beneficial effects in relation to cancer mortality and recurrence [9]. Although adherence to Mediterranean diets in cancer patients on therapy has lately been linked to a reduction in adverse side effects in studies of the bowel, breast, and prostate, no trials have been conducted in paediatric cancer patients [10,11]. As recent clinical studies have garnered interest in these dietary patterns and their components that may be responsible for their anticancer effects, they further urge investigating dietary patterns as potential nutritional interventions for paediatric patients on cancer treatment [12,13].

Polyphenols have been studied for their beneficial effects on health, particularly as cancer preventative agents where the main mechanism relies on its dual prooxidant and antioxidant action [14,15,16]. In the presence of high intracellular copper in cancer cells, polyphenols act as prooxidants when mobilizing copper ions to generate radical oxygen species, which minimizes damage to healthy cells and enables selective cancer death [15,17,18]. The clinical implication of polyphenols is their ability to play a complementary role in improving the effectiveness of cancer therapy [19,20]. Through the plethora of in vivo and in vitro studies of polyphenols with cancer cells, there has been an improved understanding of their anticancer mechanisms [21]. These include the induction of apoptosis, stimulation of immune system function, anti-inflammatory effects as well as multifaceted effects on the cellular signalling system [14,21,22]. Laboratory studies on the interactions of polyphenols as single compounds and in combinations alongside cancer therapies have provided promising results in improving treatment efficacy [23,24].

Whilst laboratory studies are limited in their extrapolation to clinical situations, clinical studies in cancer patients on treatment have revealed more positive outcomes with polyphenol combinations over single polyphenols. Polyphenol combinations as an adjuvant therapy have been associated with reducing the toxicity of chemotherapy and radiotherapy as well as been linked with additive therapeutic effects in chemotherapy for prostate [25] and colorectal cancers [19,26,27]. It is implied that polyphenols exert optimal anticancer activity when consumed in diet, due to the bioavailability and dietary relevant dosages [28] in addition to the synergistic and additive effect of polyphenol mechanisms [23,29].

The applicability and feasibility of increasing polyphenol intake as a novel dietary intervention is unknown as current research is lacking studies investigating polyphenols or the Mediterranean diet in childhood cancer patients on treatment. As it is unclear whether childhood cancer patients have a poor intake of dietary polyphenols, it is essential to determine the current polyphenol intake in children during treatment and whether it is a feasible target for future nutritional interventions. In this study, we estimated the polyphenol intake in paediatric cancer patients during treatment and explored the potential beneficial effects that a diet rich in polyphenols could have on the patient’s survival. 

## 2. Materials and Methods

### 2.1. Participants

Participants were the parents and/or primary carers of childhood cancer patients aged between 2 and 18 years, who were undergoing active cancer treatment in a major metropolitan paediatric tertiary referral hospital in Sydney, Australia. Participants were included if they were receiving active treatment for any cancer diagnosis and consuming food orally (not receiving supplementary nutrition). It was also required that the parents/caregivers of the children were sufficiently fluent in English to complete the written questionnaire. Participants were excluded if they were receiving enteral or parenteral nutrition or on maintenance therapy. 

Whilst 67 caregivers consented to participate in the study, 64 participants completed the 24-h dietary recall (24-HDR) and questionnaire, which reflects a 94% response rate. Of the 64 surveys received, five were excluded from the data analysis due to two missing questionnaire responses, one duplication, one paediatric patient being on maintenance treatment, which fell outside of the inclusion criteria, and one patient who had turned 18 during the survey and did not provide their own consent, resulting in a final sample size of 59.

### 2.2. Recruitment

Recruitment occurred between April 2016 and February 2017 after potential participants were initially identified through hospital records by a qualified dietitian familiar with oncology patients at the hospital. Participants were recruited during outpatient clinic visits or during inpatient stays by a researcher who was not involved in the care of the patient. This study is a secondary analysis of data collected for a previous study examining the dietary intake and information needs of childhood cancer patients during treatment. This study protocol was approved by the Ethics Committee at the Sydney Children’s Hospitals Network (LNR/16/SCHN/88).

### 2.3. Data Collection

A questionnaire was developed by a multidisciplinary team of researchers, medical oncologists, and paediatric dieticians. The questionnaire collected personal details of the carer as well as the child’s gender, date of birth, cancer diagnosis, and treatment and relapse status. The relapse status of the patients was collected in 2019, three years after the initial survey, where their responses were regrouped into No (no relapse) and Yes (relapse and deceased). Socioeconomic status was determined by postcode through Socio-Economic Indexes for Areas (SEIFA), where scores range from one to ten in terms of increasing relative socioeconomic advantage [30]. This was presented as low (<5), medium (5-7) and high (8-10). Body mass index (BMI) z-score percentiles were calculated by age, height, and weight measurements that were elicited from the questionnaire [31]. BMI z-scores were classified as either underweight (<5th percentile), healthy weight (5th–84th percentiles), overweight (85th–94th percentiles), or obese (>95th percentile) and used to compare across age groups in children and adolescents [32]. 

Treatment intensity of the patients was derived from their cancer diagnoses, stage, and treatment data through the validated Intensity of Treatment Rating Scale 2.0, where it ranged from 1 to 4 [33]. The scores were then regrouped into low (1 and 2), medium (3), and high (4). The paediatric Functional Assessment of Anorexia and Cachexia Therapy (peds-FAACT) was used to assess nutrition-related symptoms. This tool consists of 12 items that assess the severity of food-related symptoms in the seven days prior to completing the questionnaire [34]. The peds-FAACT subscale was used as a subjective measure of the severity of symptoms such as good appetite, sufficient food intake, bad taste, anorexia, nausea, vomiting, early satiety, and abdominal pain. The severity of the symptoms was categorized into three groups of low, medium, and high.

The dietary data were collected using a three-pass 24-h dietary recall (24-HDR), administered by a trained researcher using the following validated method. The first pass is to allow the guardian or parent to freely report all food and drink intake from the previous day uninterrupted. In the second pass, the interviewer probes for greater details on the time, type, and quantity of food or drink taken. The third and final pass is a review of everything that was recorded in order, enabling the interviewer to clarify any ambiguities and prompt for potential omissions [35]. The 24-HDR is considered one of the least biased instruments of dietary assessment and has been used in more than 50% of previous publications measuring dietary polyphenol intake [36,37,38]. This choice of dietary collection method typically relieves the burden of the responder, which facilitates a greater uptake of participants, and enables them to report all foods, whilst discouraging alteration of food intake behaviour [35].

Previous studies had primarily used the U.S. Department of Agriculture databases on polyphenol content in foods, which were limited to only flavonoid data expressed as aglycones [39,40,41]. The polyphenol content in food was assessed using the most recent Phenol-Explorer database 3.6, which compiles data on all known polyphenols, as either aglycones, glycosides, or esters depending on how they are found in foods [42]. The Phenol-Explorer database provided data on 502 polyphenol compounds in 452 plant-based foods gathered from 638 peer-reviewed articles and excluded animal foods that only contained trace amounts of polyphenols [43]. The database also divided the polyphenol content into its four main classes (flavonoids, phenolic acids, stilbenes, lignans) and “other polyphenols”, which comprised miscellaneous minor polyphenols. As there is no standardized method of assessing dietary polyphenols [36], Figure 1 depicts the collection method that was conducted according to similar publications investigating polyphenol intake in adults [43,44].

The polyphenol content of each food item was thoroughly searched in Phenol-Explorer 3.6, noting to check for the scientific names for foods or foods with different names [45]. If foods were not found in the Phenol-Explorer database, an additional literature search was conducted to find polyphenol data published since the most recent update of the database in 2013 [42].

The individual polyphenol intake of each food item was calculated by the sum of the polyphenol categories’ contents. Individual polyphenol intake from each food item was calculated by multiplying the content (mg/100 g or mg/100 mL) of each polyphenol by the daily consumption of each food (g or 100 mL), which had been calculated previously. Total polyphenol intake was calculated by the summing of all individual polyphenol intakes from the food sources reported in the dietary records per patient.

Although there are multiple extraction techniques from which the polyphenol contents of foods have been discerned in this database, this study selected chromatography to maintain consistency, as it was the most common extraction technique used in the database [42]. If data from chromatography were not available, they were obtained by chromatography-after-hydrolysis and a select few foods were measured by Folin assay. In this study, the intakes of the polyphenols are expressed as either glycosides, esters, or aglycones like other more recently published methodologies using Phenol-Explorer [43]. The Folin assay extracts the total antioxidant ability of the food as opposed to just the polyphenol content [42].

### 2.4. Data Analyses

All analyses were conducted using SPSS Statistics 25.0 (IBM Corporation, Armonk, NY, USA) The relative contributions of the food groups were calculated by the percentage of the sum of polyphenols from the specific food group over the total polyphenol intake. Individual food items were similarly assessed to evaluate the top sources of polyphenols within each specific food group [46]. The diet records were reviewed to allocate all the ingredients into food groups of fruit, vegetables, beverages, cereal, oil, nuts and seeds, seasoning, and cocoa products, according to the main ingredient’s classification in Phenol-Explorer. Of note, tomatoes and avocadoes were considered vegetables as each had been listed as a “Fruit Vegetable” in Phenol-Explorer. The contribution of each food group to the daily total intake of polyphenols was calculated from a ratio of the daily total polyphenol intake of each food group to the total polyphenol intake from all foods. Within each food group, this was replicated for the individual food sources to assess the highest contributor [46].

Dietary polyphenol intake was calculated using descriptive statistics and presented as means and errors. The contribution of each polyphenol class and food group of the total polyphenol intake was also calculated as a percentage. Demographic characteristics of the paediatric patients were calculated using descriptive statistics, presented as percentages. Due to expected changes of dietary intake across the age groups of the paediatric patients, total polyphenol intake was also presented stratified by age groups (1–3, 4–11, 12–18). These ranges were determined by the distinct age groups of toddlers, children, and adolescents as indicated by Nutrition Australia [47], which sourced their information from the Australian Dietary Guidelines [48].

Non-parametric tests (Kruskal–Wallis one-way analysis of variance) were used to compare for significant differences between groups in treatment intensity, cancer type, food-related symptoms, and total polyphenol intake. Relapse status and total polyphenol intake were compared using an independent t-test. Spearman’s correlation test was used to compare total polyphenol intake against key variables including age, years since diagnosis, BMI z-score, and socioeconomic status.

## 3. Results

### 3.1. Patient Demographics

The demographic information of the final 59 patients included in the study was gathered by the questionnaires. The mean age of the patients was 7.8 (±4.4 years) (Table 1). Just over half of the paediatric cancer patients were diagnosed with acute lymphatic leukaemia (55.9%). Most participants (90%) were undergoing chemotherapy. A total of 15 (26.3%) patients fell outside the healthy weight range, of which 3.5% were underweight, 7% were overweight, and 15.8% were obese. 

### 3.2. Polyphenol Intake

The mean and median polyphenol intakes were 173.31 ± 141.02 and 114.29 mg/day respectively (Table 2). The lower and upper percentiles are at 68.027 and 295.149 mg/day, where approximately 50% of the polyphenols were from the flavonoid family. When analysed within the distinct age groups (1–3, 4–11, 12–18 years), the means of the total polyphenol intake were 100.20, 153.44, and 273.28 mg/day and the medians were 96.49, 107.72, and 289.38 mg/day respectively. The consumed polyphenols originated from fruits (26.6%), beverages (17.98%), and cereals (16.24%) (Table 3). The beverages consumed were primarily fruit-based beverages, specifically orange, apple, and lemon juice. 

### 3.3. Socio-Demographic and Lifestyle Determinants

There was a significant, positive correlation between patient age and polyphenol intake (*p* = 0.001). There was no significant correlation between time since diagnosis, BMI z-score, types of cancer, treatment intensity, and total daily polyphenol intake. Early satiety and taste changes were the most common food-related symptoms reported by the patients, though the overall incidence of food-related symptoms was low (Figure 2). There were no significant associations found between any of the food symptoms and polyphenol intake. Twelve percent of patients had relapsed or died three years after the study recruitment. There was also no difference between the relapse status and polyphenol intake but a general trend towards higher polyphenols in the non-relapse groups.

## 4. Discussion

New research is suggesting that dietary polyphenols may improve treatment outcomes in cancer in adult patients. There is a dearth of research assessing polyphenols and children undergoing treatment for cancer. Our study shows that childhood cancer patients have low intakes of polyphenols during their cancer treatment, particularly when compared to equivalent age groups of toddlers, children, and adolescents. Fruits, beverages, and cereals were the main sources of polyphenols for childhood cancer patients during treatment. Within these food groups, polyphenols were predominantly sourced from the children’s intake of apples, fruit juice, and bread respectively. This study offers insight into the potential of polyphenols as a focus in future nutritional adjuvant therapies in paediatric cancer patients.

There is a lack of normative data for polyphenol intake in children. Five studies have assessed the total polyphenol intake in children without cancer or other comorbidities. Polyphenol intake in the non-cancer population from the United Kingdom and Argentina ranges from 266 mg/day to 412 mg/day [44,49]. This mean polyphenol intake is higher than the mean intake in childhood cancer patients of 173.31 mg/day. The median intake of polyphenols in the non-cancer population in Europe and Poland ranged from 171 mg/day to 152 mg/day [50,51,52], which again was higher than the median intake of 114.29 mg/day in childhood cancer patients. Across all age groups, there was an approximately 100 mg/day lower intake of polyphenols in childhood cancer patients compared with the non-cancer paediatric population. Taken together, these results would suggest that childhood cancer patients have a low intake of polyphenols compared with children without cancer. 

The finding of a low polyphenol intake may reflect the dietary difficulties typically faced by paediatric cancer patients as the adverse effects of treatment often negatively impact their eating experience [2]. Paediatric cancer patients experience behavioural changes towards food and physiological changes resulting from cancer and the treatments, placing them at a greater risk of malnutrition throughout treatment [53]. Interestingly there were no significant associations between the polyphenol intake and food symptoms of cancer treatment, and many patients were not experiencing severe dietary symptoms. Previously, concerns for adverse food symptoms arising from treatment and cancer intensity as well as the survival outcomes from therapy have pushed prioritizing a high energy diet in childhood cancer care above a diet of good nutritional quality [4,54,55]. Studies in adults with cancer have suggested that increasing the intake of polyphenols during treatment may contribute to a reduction in symptoms from the toxicity of the cancer treatment and limiting cancer progression [14,19,27]. As cancer treatment in children has been linked with the depletion of their bodies’ stores of antioxidants, improving the dietary intake of children with cancer with a focus on polyphenols may have the potential to reduce their treatment-related side effects [55]. 

To promote paediatric cancer patients’ intake of polyphenols, interventions should recognize key foods and food groups that are rich in polyphenols and consider dietary patterns that would best suit the patients’ needs [56]. There is greater benefit to consuming whole foods over supplements, as it achieves a synergistic effect from a complex combination of natural polyphenols and optimizes health from having more fruits and vegetables [57,58,59]. Food groups rich in polyphenols, including fruits, vegetables, whole grains, legumes, seasonings, and olive oil, have a high polyphenol content [60]. Foods that are more appealing to children such as dark chocolate as an alternative to milk chocolate could be encouraged, due to their relatively high polyphenol content to weight ratio. The intake of milk chocolate products over dark chocolate products attributed to the low polyphenol contribution from cocoa products in this study. Due to the varied bioavailability and absorption of different polyphenols in the body, it may be more feasible to base dietary interventions on polyphenol-rich dietary patterns such as the Mediterranean diet [14,59]. The Mediterranean diet has been well studied for its role as an adjuvant in the treatment of adult cancers [9,10,11,61]. The advantage of encouraging a Mediterranean diet is that it is a dietary pattern associated with nutritional adequacy in children [62]. It has the potential to be accepted by children undergoing treatment for cancer and still maintain the mechanistic effect of polyphenols [8]. Future dietary interventions that focus on polyphenol-rich foods may consider using the Mediterranean diet as a framework.

In this study, there was a notable lack of polyphenol contribution from vegetables. This poor intake of vegetables reflects a trend in the diets of childhood cancer patients undergoing treatment, indicating a common target for dietary interventions for this population [63,64]. A high vegetable intake is vital in providing the necessary nutrients unavailable from other food groups, and is associated with reducing the risk of developing cancer and other chronic health conditions [65,66]. However, current nutritional recommendations for paediatric cancer patients focus on mitigating malnutrition by preventing weight loss rather than regulating the food groups from which the calories are sourced [2,55]. A healthy diet should be recommended for childhood cancer patients combining high-energy and high-quality food. 

Future nutritional interventions should optimize an intake of diverse polyphenols through a dietary pattern that promotes healthy consumption of fruits and polyphenols, like the Mediterranean diet. There is strong possibility that adopting a healthy dietary pattern in paediatric cancer patients may reduce the toxic side effects of cancer therapy, potentially improving treatment outcomes. Whilst no firm recommendations can be made as of yet for childhood cancer patients, it is worth noting the characteristic polyphenol-rich foods in the Mediterranean diet that are suitable for children: vegetables, fruits, nuts, olive oil and herbs, and spices [67]. Like all diets, it is important to consume foods as a variety and in moderation. Developing optimal dietary habits throughout treatment also sets the foundations for a healthier diet as an adult, minimizing the increased risk of developing adverse chronic conditions as a cancer survivor. Further research is required to develop a framework for a polyphenol-rich diet tailored to paediatric cancer patients, before integrating the intervention into the long-term management of paediatric cancer.

### Strengths and Limitations

The present study has several strengths, particularly in its use of the Phenol-Explorer 3.6 database, the standardized use of 24-HDR, and developing a stringent methodology to investigate polyphenols in children’s diets. This study utilized the most updated and comprehensive Phenol-Explorer database whilst other descriptive studies used smaller and local databases to document the intake of polyphenols, which may be more suitable due to cultural dietary differences [42]. This study is also the first exploratory study of polyphenols in paediatric cancer patients on treatment, in a period in research where dietary polyphenols are of interest in cancer research.

However, this study also has a few limitations. The major limitation is represented by the sample size of 59, an acceptable size compared to other dietary studies in childhood cancer patients on treatment, which do not typically vary above 100 [68,69]. Future research should still seek to recruit more participants to improve the power of the analysis as the current study size limits the power of the correlation analyses, resulting in insignificant associations. This will also improve the ability to stratify food group contributions to total polyphenol intake by age group, enabling a more tailored understanding of each age group’s characteristic intakes. This study’s participants were also recruited in a single hospital located in a developed country, limiting extrapolations to populations with dissimilar characteristics.

As this was a secondary analysis, the dietary recall was not specific to collecting polyphenols. To increase accuracy in future studies measuring polyphenols, the 24-HDR should be adapted to include questions tailored to screen for sources of polyphenols, such as clarification on the oil used in cooking and the composition of herbal tea. Cooking methods should also be noted whenever possible in order to utilize the retention factors of Phenol-Explorer 3.6. Additionally, future studies should use multiple 24-HDRs over consecutive days, which would more accurately assess the habitual dietary intake of polyphenols, particularly in smaller sample sizes.

Furthermore, whilst Phenol-Explorer 3.6 is currently the most updated database for polyphenols since 2015, there may still be missing and incomplete composition data for foods with polyphenols due to lack of research in certain food items [42]. This results in a potential underestimation of total polyphenol intake, which had been mitigated by our own search through the literature for additional studies not yet included in the database [70]. This additional literature search for new studies looking into polyphenol composition in different foods should be included in the methodology of other similar studies in the future.

Whilst this study does look at the approximate dietary intake of polyphenols, it does not ascertain the level of polyphenols to achieve therapeutic effects in childhood cancer patients. This topic should be an area of interest, which would help shape and guide future nutritional guidelines in terms of determining an appropriate dietary intake of polyphenol-rich foods.

## 5. Conclusions

Our study has revealed a poor intake of polyphenols in paediatric cancer patients and an opportunity for a new targeted nutritional intervention. Ultimately, the success of a dietary therapy must consider many aspects of paediatric cancer care, including the nutritional framework it is based on as well as its integration into paediatric cancer management. To develop an effective dietary framework that improves the efficacy of cancer therapy, more research is needed to appreciate the most applicable foods and dietary patterns in paediatric cancer therapy. In the future, this would also require trials in children with cancer to determine the most effective influences for successful adherence. Introducing a new dietary pattern for paediatric cancer patients would require careful consideration for the individual nutritional challenges experienced, tailored accordingly by the medical oncologist and dieticians involved in their management. Further investigation into larger studies of paediatric cancer patients will facilitate the steps in assessing the value of polyphenol-rich dietary patterns in future nutritional interventions.

## Figures and Tables

**Figure 1 nutrients-11-02835-f001:**
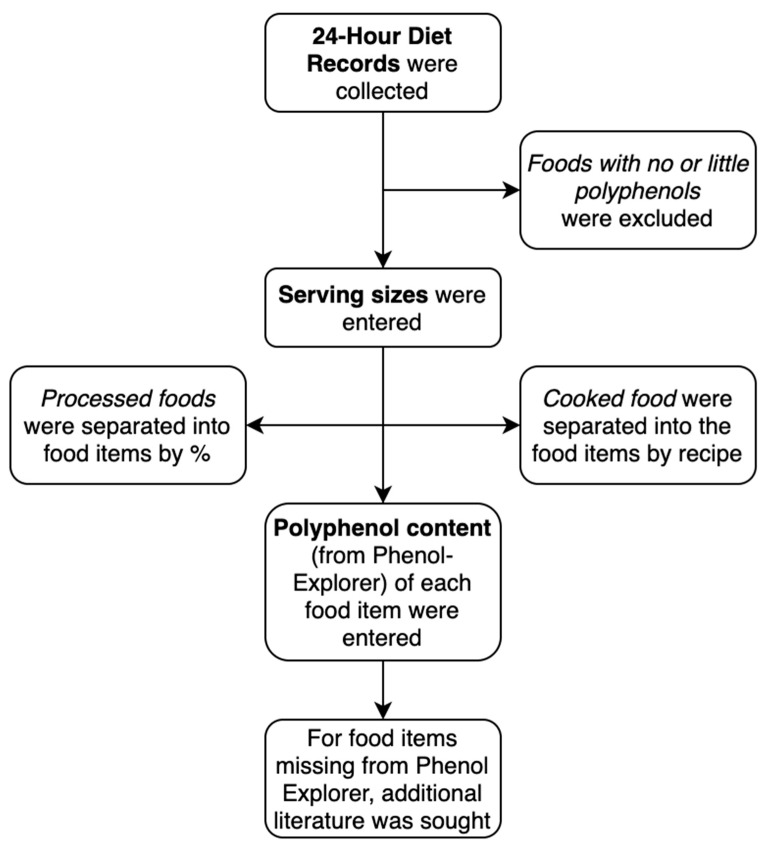
Method for the extraction of total polyphenol intake from the 24-HDR (24-h dietary recall).

**Figure 2 nutrients-11-02835-f002:**
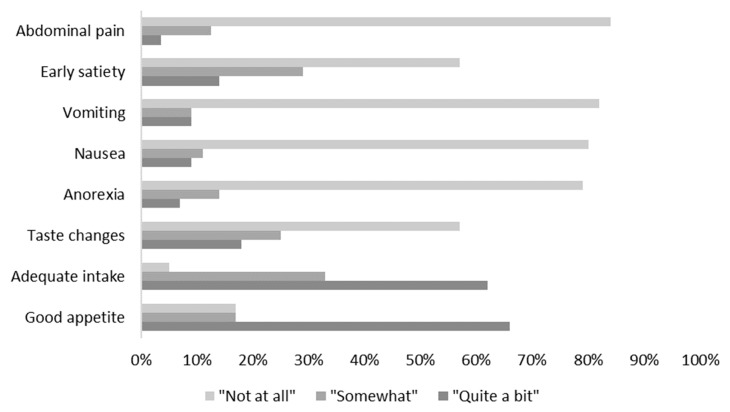
Percentage of patients experiencing food symptoms.

**Table 1 nutrients-11-02835-t001:** General demographic characteristics of the present study.

Characteristics	N (%)	Mean (Range, SD)
Age at time of survey (nearest year)		7.8 (2–16, 4.4)
Years since diagnosis		5.9
Age at diagnosis		
Gender		
Female	24 (40.7%)	
Male	35 (59.3%)	
Body mass index (BMI) z-score ^1^		17.4 (12.7–23.2)
Obese	9 (15.8%)	
Overweight	4 (7%)	
Healthy	42 (73.7%)	
Underweight	2 (3.5%)	
Socioeconomic status		
Low	9 (15.2%)	
Medium	8 (13.6%)	
High	42 (71.2%)	
Diagnosis		
Acute lymphoblastic leukaemia	33 (55.9%)	
Acute myeloid leukaemia	1 (1.7%)	
Brain cancer (e.g., Medulloblastoma, Glioma)	4 (6.8%)	
Hodgkin’s lymphoma	4 (6.8%)	
Neuroblastoma	2 (3.4%)	
Non-Hodgkin’s lymphoma (including Burkitt’s lymphoma)	4 (6.8%)	
Wilms’ tumour	4 (6.8%)	
Sarcoma of the bone (e.g., Osteosarcoma)	2 (3.4%)	
Soft tissue sarcoma	1 (1.7%)	
Other	4 (6.8%)	
Relapse ^2^		
Yes	7 (12.1%)	
No	51 (87.9%)	
Treatment received		
Surgery	13 (22%)	
Chemotherapy	54 (91.5%)	
Radiotherapy	10 (16.9%)	
Bone marrow transplant/stem cell transplant	5 (8.5%)	
Other	5 (8.5%)	
Treatment Intensity		
1	2 (3.4%)	
2	28 (47.5%)	
3	20 (33.9%)	
4	9 (15.3%)	

^1^*n* = 57, ^2^
*n* = 58.

**Table 2 nutrients-11-02835-t002:** Intakes of different classes of polyphenols and total polyphenol intake.

Polyphenol Classes	Mean ± SD (Mg/Day)	Median (Mg/Day)	Range (Mg/Day)
Flavonoids	99.85 ± 116.42	51.87	0.00–541.72
Phenolic acids	43.70 ± 36.94	31.49	1.64–150.45
Lignans	13.33 ± 21.51	4.24	0.00–120.69
Stilbenes	0.25 ± 0.75	0.00	0.00–3.53
Other	15.09 ± 20.56	8.02	0.00–90.31
Total polyphenol intake ^1^	173.31 ± 141.02	114.29	3.42–665.54

^1^ Mean of total polyphenol intake does not match the sum of the mean of polyphenol classes due to missing data for the polyphenol classes when using the Folin Assay extraction technique.

**Table 3 nutrients-11-02835-t003:** Contributions of polyphenols from each food group and food sources.

Food Groups	Average Polyphenol Intake (Mg/Day)	Relative Contribution (%)	Top 3 Common Food Sources
Fruit	44.74	26.63	Apple, banana, strawberry
Vegetable	38.35	9.34	Tomato, potatoes, cucumber
Cereal	22.39	16.24	Whole meal bread, rice, bread (other)
Oils	3.97	4.64	Olive oil (virgin, extra-virgin), vegetable oil
Beverages	50.49	17.98	Orange juice, apple juice, lemon juice
Nuts and seeds	3.96	1.56	Primarily found in muesli and seeded bread
Seasoning	3.04	1.38	Garlic, soy sauce, basil
Cocoa products	5.13	2.23	Cocoa, hot chocolate

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
