# Peer review of "Poor Dietary Polyphenol Intake in Childhood Cancer Patients"

_nutrients, 2019, doi:10.3390/nu11112835_

Round 1

Reviewer 1 Report

Paper titel: Poor dietary polyphenol intake in childhood cancer patients described by Liu et al.

In this study, Liu et al. demonstrates polyphenol-rich diets like the Mediterranean diet may play a role in improving the outcomes of people cancer therapy. This paper is really interested because nowadays still childhood cancer patients  have a poor intake of dietary polyphenols, it is essential to determine the current polyphenol intake in children during treatment and whether it is a feasible target for future nutritional interventions. In this study, Authors consequently examine the polyphenol intake in pediatric cancer patients during treatment and to understand the potential beneficial effects that a diet rich in polyphenols could have on patient’s survival.

Data are organized in a scientific order.

Additionally comments:

The aim of this study show present much more clear.

How many patients was in this study. The population was big or not ? shoud be described details in M section not in 3.1 section.

Describe more information about some limitation of this study. How we should eliminated this ?

Please, specify how many polyphenolic compounds should be consumed and what kind of compounds to achieve strong therapeutic effects?

Please, specify how many polyphenolic compounds should be consumed and what kind of compounds to achieve strong therapeutic effects?
What products (specifically) should be in the diet of such people.
Authors should add some information compare to literature and discussion on how much diet can counteract this disease?

Additionally in all paper but especially in R&D section should more be disused obtain results.

After correction this point paper should be again evaluated.

Reviewer 2 Report

Dear authors many thanks to have proposed this study for publication. The Polyphenols are natural compounds capable of interfering with the inflammatory pathways of several in vitro model systems. Several studies have described the beneficial effects of plant-derived polyphenols as natural ligands that are able to reduce inflammation, with some inhibiting production of TNFα from cell lines of different origins in both in vitro and in vivo animal models.  Compared with the effects of polyphenols in vitro, the possible functions in vivo and in humans remain unknown due to the absence of validated in vivo biomarkers and long-term studies failing to demonstrate effects with a clear mechanism of action, appropriate sensitivity and specificity or efficacy.

I think that this study was very well conducted. Methods and materials were well described and developed. I completely agree with all the strengths and limitations listed, that were well developed and discussed.

Reviewer 3 Report

Main comments:

Line 113, What does it mean “three-pass 24-hour dietary recall”? It is based on multiple 24HDRs (n=3)? Were they 3 consecutive days? Please give more details and include more information in the abstract. One 24HDR is not enough to estimate accurately the habitual intake of polyphenols in a small population. Line 140-145.Please indicate if the intake was computed as aglycones or as they are found in nature (glycoside and esters). You cannot combine data from “chromatography” and “chromatography after hydrolysis” if you are presenting data as glycosides. This is a very important issue that you need to clarify and explain in more detail. Line 170, why did you use a Pearson’s correlation when you dietary data is not normally distributed. The results on cocoa products seem very low, 5mg polyphenol/d means that these children consume less than 1 portion of dark chocolate per week. Table 2 and 3 should be done stratified by age. Lines 219-228, it is very important to compare the results by age group, since the intake varies between 68 and 295mg/d in “toddlers” and “adolescents”, respectively. Mean intake in the entire population is based in the age of the participants

Minor comments:

Line 36-37, polyphenol intake in Mediterranean diets is lower than in Nordic European diets (Zamora-Ros, R., et al., Dietary polyphenol intake in Europe: the European Prospective Investigation into Cancer and Nutrition (EPIC) study. European Journal of Nutrition,2016. 55(4): p. 1359-1375.) , because coffee and tea are the main contributors. Line 72, please change “examine” for “estimate”. Line 136 (mg/g or mg/100ml) -> (mg/100g or mg/100ml) Line 167 Non-parametric tests (Kruskal-Wallis one-way analysis of variance) “were” used… Line 181, (ALL)??? Please check the references. For example: Refs 15 & 16, the volume and pages are missing. Ref 40, the journal is missing. Please change ref 39 by : Neveu V, Perez-Jiménez J, Vos F, Crespy V, du Chaffaut L, Mennen L, Knox C, Eisner R, Cruz J, Wishart D, Scalbert A. Phenol-Explorer: an online comprehensive database on polyphenol contents in foods. Database (Oxford). 2010;2010:bap024.

Round 2

Reviewer 1 Report

Present manuscript after correction is siutable for print in this journal. 

Reviewer 3 Report

I do not have further comments.